# TRACE: LEARNING TO COMPUTE ON GRAPHS

## ABSTRACT

Learning to compute—the ability to model the functional behavior of a computational graph—is a fundamental challenge for graph representation learning. Yet, the dominant paradigm is architecturally mismatched for this task. This flawed assumption, central to mainstream message passing neural networks (MPNNs) and their conventional Transformer-based counterparts, prevents models from capturing the position-aware, hierarchical nature of computation. To resolve this, we introduce **TRACE**, a new paradigm built on an architecturally sound backbone and a principled learning objective. First, TRACE employs a Hierarchical Transformer that mirrors the step-by-step flow of computation, providing a faithful architectural backbone that replaces the flawed permutation-invariant aggregation. Second, we introduce **function shift learning**, a novel objective that decouples the learning problem. Instead of predicting the complex global function directly, our model is trained to predict only the *function shift*—the discrepancy between the true global function and a simple local approximation that assumes input independence. We validate this paradigm on electronic circuits, one of the most complex and economically critical classes of computational graphs. Across a comprehensive suite of benchmarks, TRACE substantially outperforms all prior architectures. These results demonstrate that our architecturally-aligned backbone and decoupled learning objective form a more robust paradigm for the fundamental challenge of learning to compute on graphs.

## 1 INTRODUCTION

Computational graphs provide a fundamental abstraction for modeling computation. As directed graphs of nodes representing operations and variables, they capture the flow of computation and are crucial in domains ranging from control data flow graphs (CDFGs) in software engineering to electronic circuits in hardware design. Accurately modeling the computational behavior of these graphs is therefore a critical enabler for high-impact applications, including performance prediction (Chen et al., 2024; Xie et al., 2022; Zhang et al., 2020; Mendis et al., 2019), verification (Li et al., 2023; Selsam et al., 2018; Zhang et al., 2021), and optimization (Zuo et al., 2023; Mirhoseini et al., 2021).

This need for functional modeling has driven recent work in graph representation learning, with approaches largely divided into two families: message passing neural networks (MPNNs) and Graph Transformers. These models have been increasingly applied to capture the functionality of diverse computational graph modalities, with a significant body of work focusing on the particularly challenging domain of hardware design, including Register Transfer Level (RTL) graphs (Fang et al., 2025c;b), And-Inverter Graphs (AIGs) (Shi et al., 2023; 2024; Zheng et al., 2025; Khan et al., 2025; Wu et al., 2025; Wang et al., 2024; Liu et al., 2024), and post-mapping (PM) netlists (Shi et al., 2025b). MPNNs provide a general framework (Gilmer et al., 2017) based on a paradigm of aggregating and updating node features:

$$\mathbf{x}_i' = \gamma\left(\mathbf{x}_i, \underset{j \in \mathcal{N}(i)}{\square} \varphi\left(\mathbf{x}_i, \mathbf{x}_j, \mathbf{e}_{j,i}\right)\right), \tag{1}$$

where $\square$ denotes a permutation-invariant aggregator (e.g., sum, mean, max) and the update function $\gamma$ and message function $\varphi$ are differentiable functions. In contrast, Transformer-based methods either flatten the graph into a sequence for global self-attention (Fang et al., 2025c;b), or incorporate graph structure via attention masks (Shi et al., 2024; Zheng et al., 2025; Fang et al., 2025a).

Figure 1: The architectural failure of MPNNs on computational graphs. **Left:** The permutation-invariant aggregation in MPNNs cannot distinguish between ordered inputs (e.g., A,B vs B,A), yielding the same incorrect embedding for a position-aware operator like MUX. **Right:** Our approach processes inputs as an ordered sequence, enabling position-awareness and capturing operator-specific interactions.

Despite their success on general graph domains, we argue that these architectures are fundamentally ill-suited for computation due to a **fundamental mismatch** between their properties and the nature of computational graphs (as shown in Fig. 1):

1. **MPNNs Fail to Model Input Interactions:** The core architectural flaw of MPNNs is that their message functions model interactions between an operator and only a single input at a time. These independent messages are then combined by a permutation-invariant aggregator. This two-stage process makes it architecturally impossible to capture the intricate, operator-specific relationships between multiple inputs and, as a direct consequence, to model position-aware operators like the MUX (multiplexer), where input order is critical (i.e., MUX(S, A, B) ≠ MUX(A, S, B)).

2. **Vanilla Transformers are Hierarchy-Agnostic:** Fully-connected Transformers flatten the graph into a sequence, destroying the explicit hierarchy and connectivity crucial for modeling computation. For an expression like $y = f(g(a,b), c)$, this architecture fails to capture the intermediate dependency on $g(a,b)$ and cannot guarantee a correct computational trace.

3. **Edge-masked Transformers Inherit MPNN Limitations:** Edge-masked transformers, while respecting connectivity, often reduce to GAT-like attention mechanisms. This is functionally a weighted sum, which still inherits the fundamental limitations of the message-passing paradigm, failing to model the precise, non-linear interactions required by logical and algebraic operators.

These architectural flaws highlight a failure to model the step-by-step flow of computation. Beyond this, however, a deeper challenge lies in capturing global function that emerges from the graph's overall topology. Even with simple components (e.g., AND and NOT gates in AIGs), a graph's overall functionality can become highly complex due to reconvergent dependencies. For instance, consider $c = a \wedge b$ with $a = x \wedge y$ and $b = y \wedge z$, where $x, y, z \sim \mathcal{B}(p)$ (a Bernoulli distribution with parameter $p$). Locally, $a, b \sim \mathcal{B}(p^2)$, and ignoring reconvergence, one would predict $c \sim \mathcal{B}(p^4)$. However, since $y$ appears in both $a$ and $b$, $a$ and $b$ are correlated, shifting the true distribution to $c \sim \mathcal{B}(p^3)$. This *function shift*—from $\mathcal{B}(p^4)$ to $\mathcal{B}(p^3)$—demonstrates how dependencies fundamentally alter functional behavior. Previous works that supervise directly on a node's final global function implicitly bundle this effect into the embeddings, making it difficult for the model to distinguish true functional dependencies from spurious correlations.

Our work addresses these challenges with **TRACE**, a Transformer for Reasoning about Algebraic and Computational Expressions, which presents a two-fold solution. First, to resolve the local architectural flaws of prior models, TRACE employs a hierarchical Transformer. Inspired by prefix notation, we represent each computation step as an ordered sequence, [operator, input_1, input_2, ...], which is processed by a Transformer encoder with positional encoding (Figure 1). By applying this process recursively according to the graph's logical dependencies, TRACE learns a faithful, position-aware representation of each computational step. Second, to capture the global function, we introduce **function shift learning**, a novel objective that explicitly models the discrepancy between local and global functions. This allows the model

to disentangle a node's intrinsic behavior from the contextual effects imposed by the wider graph topology.

We demonstrate the effectiveness of TRACE on electronic circuits—a particularly challenging and representative class of computational graphs. Our experiments span a broad spectrum of circuit modalities (RTL, AIG, and PM netlists) and tasks (contrastive and predictive) across several standard benchmarks, including ITC (Corno et al., 2002), OpenCores (Albrecht, 2005), ISCAS '89 (Brglez et al., 1989), ForgeEDA (Shi et al., 2025a), and DeepCircuitX (Li et al., 2025). The results are unequivocal: TRACE consistently and substantially outperforms prior approaches across all settings. This establishes TRACE not only as a new state of the art for circuit analysis, but as a more robust and architecturally sound paradigm for learning on computational graphs.

## 2 BACKGROUND

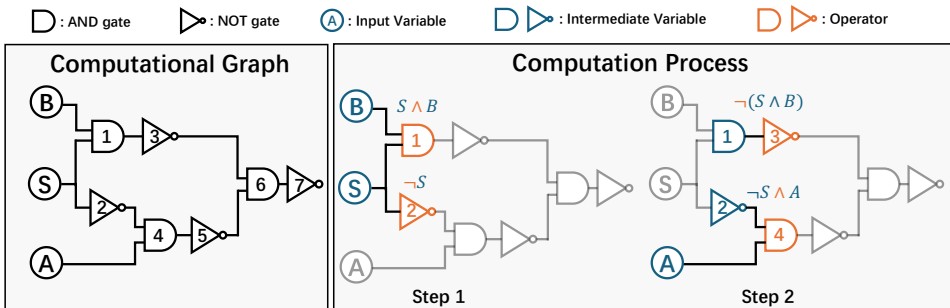

Figure 2: Illustration of a Computational Graph and Its Computation Process. This figure demonstrates the dual role of nodes within a computational graph. **Step 1** shows nodes 1 and 2 functioning as operators to compute the expressions $S \wedge B$ and $\neg S$, respectively. As the computation progresses to **Step 2**, these nodes transition to representing the intermediate variables that hold the results of these operations in step 1, which are then passed to subsequent operators for further computation.

### 2.1 COMPUTATIONAL GRAPHS

In our work, computational graphs are defined as directed graphs where nodes represent either input variables or operators, and edges signify the flow of data. As depicted in Figure 2, this structure allows a single node to serve a dual role: it can be an operator receiving data from its source nodes and an intermediate variable whose output is consumed by other operators. Specifically, for any directed edge from a source to a target node, the target always represents an operator, while the source represents a variable—either an initial input variable or a temporary intermediate variable resulting from a previous operation. This representation effectively models the data dependencies and computational flow within the system. We focus on three types of computational graphs from the front-end of the electronic design automation (EDA) flow: Register-Transfer Level (RTL) graphs, And-Inverter Graphs (AIGs), and Post-Mapping (PM) netlists. The primary distinction among these graph types lies in their operators. AIGs are composed of a minimal set of basic operators (AND and NOT gates), while RTL and PM netlists feature more complex, higher-level operators, such as multiplexers where $\texttt{MUX}(\texttt{S},\texttt{A},\texttt{B}) = (S \wedge B) \vee (A \wedge \neg S)$. For a more detailed description of the computational graphs used in this paper, please refer to Appendix B.

A notable property of computational graphs, and one that differentiates them from general graphs, is the distribution of node in-degrees. Unlike the often long-tailed distribution found in general graphs, the in-degree of a node in a computational graph is directly related to its operator type, leading to a stable and predictable distribution. This inherent property, detailed further in Appendix D, contributes to the substantially reduced padding overhead of our proposed model.

### 2.2 MESSAGE PASSING NEURAL NETWORKS

Message Passing Neural Networks (MPNNs) are a dominant architectural paradigm for function learning on circuit graphs. These models can be broadly categorized into two types: synchronous and asynchronous. Synchronous MPNNs (Wu et al., 2023a; Liu et al., 2024; Wu et al., 2025;

Deng et al., 2024) process all message-passing updates in parallel, a strategy designed for computational efficiency. In contrast, asynchronous MPNNs (Li et al., 2022; Shi et al., 2023; 2025b; Khan et al., 2025; Wang et al., 2024) mimic the logic simulation process by updating node representations sequentially, following a topological order. This approach aims to capture effective functional representations by emulating the data flow. However, as illustrated in Figure 1, both synchronous and asynchronous MPNNs rely on the conventional message passing paradigm (Gilmer et al., 2017), which fundamentally struggles to capture the operator-specific and position-aware interactions among input variables.

## 2.3 GRAPH TRANSFORMERS

Despite the widespread use of MPNNs, they suffer from inherent limitations, including difficulty in capturing long-range dependencies and susceptibility to issues like over-smoothing (Li et al., 2018) and over-squashing (Alon & Yahav, 2020). These drawbacks have motivated a shift towards Graph Transformers, which leverage global attention mechanisms to address these limitations. Existing graph transformer models can be classified into two main categories: fully-connected transformers and edge-masked transformers.

Fully-connected transformers, such as Rampášek et al. (2022); Wu et al. (2023c;b) for general graphs and Fang et al. (2025c;b) for circuit graphs, treat the graph as a flattened sequence of nodes. While this enables global self-attention, it inadvertently leads to a loss of the critical hierarchical structure and intermediate dependencies that are inherent to computational graphs. In contrast, edge-masked transformers, including DeepGate3 (Shi et al., 2024), DeepGate4 (Zheng et al., 2025), and Net-TAG (Fang et al., 2025a), integrate the graph's topology by using the adjacency matrix to mask the attention mechanism. However, this approach often reduces the model to a Graph Attention Network (GAT)-like attention mechanism, inheriting the limitations of traditional MPNNs.

## 3 METHOD

### 3.1 OVERVIEW

Our framework learns circuit functionality using a Hierarchical Transformer backbone trained with two objectives: a predictive task and a contrastive task. The Hierarchical Transformer, detailed in Section 3.2, provides an architectural backbone that mirrors the circuit's computational flow, replacing the conventional message-passing paradigm. For the predictive task, we introduce Function Shift Learning (FSL) in Section 3.3, a novel objective that explicitly models the discrepancy between a circuit's local and global functions.

### 3.2 HIERARCHICAL TRANSFORMER

To address the limitations of MPNNs and Graph Transformers discussed in Section 1 and 2, we propose a new paradigm for encoding the computational graph, with a position-aware Hierarchical Transformer that enables the operator-specific interaction among input variables, faithfully mirroring the computational process illustrated in Figure 2.

A computational graph $\mathcal{G} = (\mathbf{V}, \mathbf{E})$, consists of primary input (PI) nodes, which have an in-degree of zero, and operator nodes, whose in-degree is determined by their type. We begin by computing the logic level of each node in topological order[1] as follows:

$$level(v) = \begin{cases} 0 & \text{if } v \text{ is PI} \\ 1 + \max_{(u,v) \in \mathbf{E}} level(u) & \text{otherwise} \end{cases} \tag{2}$$

The logic level defines the computational dataflow through the graph. Inspired by asynchronous MPNNs, we process nodes level by level. First, Primary Input (PI) nodes (level 0) are initialized with their input distribution, e.g., $\mathcal{B}(p)$, a Bernoulli distribution with parameter $p$. Then, for the nodes

---

[1]To handle cyclic graphs, such as sequential AIGs, we follow Khan et al. (2025) by treating all flip-flops or registers as pseudo Primary Inputs (PIs) and removing the feedback edges to compute the logical level.

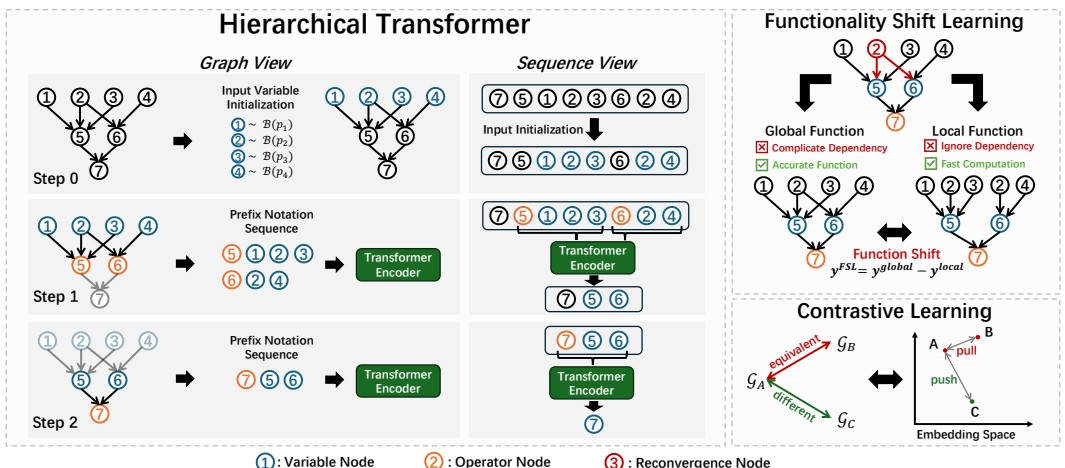

Figure 3: Overview of our proposed framework. **Left:** A circuit graph, represented in both a graph view and its equivalent prefix notation, is encoded by a Hierarchical Transformer to model the computation process. **Right:** For predictive tasks, we introduce Function Shift Learning (FSL). Instead of directly regressing the global function, the model captures the difference between the global and local functions: $y^{FSL} = y^{global} - y^{local}$. For contrastive tasks, we pull embeddings of equivalent circuits closer while pushing apart those of functionally different circuits.

$v_0^k, v_1^k, \ldots, v_{n_k}^k$ at subsequent level $k > 0$, we gather their respective sets of direct predecessors, $\mathcal{N}(v_0^k), \ldots, \mathcal{N}(v_{n_k}^k)$, where each set is defined as $\mathcal{N}(v_i^k) = \{u_j \in \mathbf{V} \mid (u_j, v_i^k) \in \mathbf{E}\}$.

Inspired by prefix notation, we represent each computation at level $k$ as an ordered sequence, $\mathbf{v}_i^k$. This sequence is constructed by placing the *operator* $v_i^k$ at the head, followed by its *ordered input nodes* from $\mathcal{N}(v_i^k)$:

$$\mathbf{v}_i^k = [v_i^k, u_{j_1}, u_{j_2}, \ldots, u_{j_{|\mathcal{N}(v_i^k)|}}], \text{ where } u_{j_1}, \ldots, u_{j_{|\mathcal{N}(v_i^k)|}} \in \mathcal{N}(v_i^k) \tag{3}$$

To model operator-specific interactions among inputs, we apply a Transformer encoder to the sequence $\mathbf{v}_i^k$ augmented with positional encodings. The updated embedding for the operator node $v_i^k$ is taken from the Transformer's output corresponding to the first token:

$$v_i^k = \text{Transformer}(\mathbf{v}_i^k + \mathbf{pos})[0], \tag{4}$$

where $\mathbf{pos}$ represents the positional encodings. The input sequence $\mathbf{v}_i^k$ is composed of the initial embedding of the operator $v_i^k$ (e.g., a one-hot vector of its type) and the embeddings of its input nodes $\{u_{j_m}\}$, which have been computed in previous steps. After updating, the embedding of $v_i^k$ now represents the result of the computation at this node, i.e. an intermediate variable node.

This paradigm offers several key advantages. First, positional encodings enable position-aware aggregation, a critical feature for order-dependent operations, which contrasts with the permutation-invariant nature of standard message-passing schemes. Second, the self-attention mechanism facilitates rich interactions among all source nodes $[u_{j_1}, \ldots, u_{j_{|\mathcal{N}(v_i^k)|}}]$, faithfully mirroring the computational dataflow of an operator, as illustrated in Figure 2. Finally, the sequence length for the Transformer depends only on a node's in-degree. Unlike general graphs, which often have a long-tailed degree distributions that incur significant overhead, circuit graphs have small, tightly bounded in-degrees. This inherent property, detailed further in Appendix D, contributes to the substantially reduced padding overhead of our proposed model, as we discussed in Section 2 and Appendix D.

**An Alternative View: Hierarchical Transformer on Prefix Notation**   Our proposed method can also be interpreted as a Hierarchical Transformer operating on prefix notation, in a way that follows the logical dependencies inherent in the computational graph. This perspective highlights how the model processes the graph by mirroring the step-by-step evaluation of an expression. A computational graph can be converted into a prefix notation string through a pre-order traversal on its reversed

edges. As illustrated in Figure 3, our proposed Hierarchical Transformer, unlike methods that simply flatten the graph into a single sequence, inherently preserves the nested, hierarchical structure of dependencies within each computation step, which allows it to align naturally with the actual flow of computation. This view also underscores the generalizability of our approach to other sequence-based problems, such as hardware model-checking, where the problem is often represented in Btor2 format (ArminBiere et al., 2018), a prefix-style representation of bit-vector formulas.

### 3.3 Function Shift Learning on Predictive Task

Logic-1 probability prediction is a task widely studied in prior works (Shi et al., 2023; Khan et al., 2025; Shi et al., 2024; Zheng et al., 2025; Shi et al., 2025b; Liu et al., 2024), as it serves as a key indicator of a model's ability to capture circuit functionality. The logic-1 probability corresponds to the *global function* of a circuit, as defined in Definition 1. This function can be highly complex due to reconvergent dependencies (see Section 1), and computing it directly requires enumerating the joint distribution of all inputs, which incurs an exponential cost of $O(2^k)$.

**Definition 1** (Global Function). *Given an operator $\phi$, input variables $\mathbf{x} = [x_1, x_2, \ldots, x_k]$ and its distribution $\mathcal{D}$, the global function is defined as $y_\phi^{global} = \mathbb{E}_{\mathbf{x} \sim \mathcal{D}}[\phi(x_1, x_2, \ldots, x_k)]$.*

By ignoring the dependencies among input, i.e. by assuming they are independent, we can derive the *local function*, which is formally stated in Definition 2. Although this approximation is computationally efficient with an $O(1)$ complexity, it fails to capture the true function of the circuit.

**Definition 2** (Local Function). *Given an operator $\phi$, input variables $\mathbf{x} = [x_1, x_2, \ldots, x_k]$ and its distribution $\mathcal{D}$, the local function is defined as $y_\phi^{local} = \phi(\mathbb{E}_{\mathbf{x} \sim \mathcal{D}}[x_1], \mathbb{E}_{\mathbf{x} \sim \mathcal{D}}[x_2], \ldots, \mathbb{E}_{\mathbf{x} \sim \mathcal{D}}[x_k])$.*

Building on these properties, we propose to learn the *function shift*: $y_\phi^{FSL} = y_\phi^{global} - y_\phi^{local}$, which measures the discrepancy between the local function (Definition 2) and the true global function (Definition 1). This formulation decouples the global function into two components: a simple local function and a function shift. Rather than predicting the complex global function directly, our model is trained to predict only the function shift. This isolates the complex contextual effects caused by reconvergence, allowing the global function to be reconstructed by simply combining the predicted shift with the local function.

**Training Stage.** During the training stage, the ground-truth global and local functions for each node can be pre-computed from the training data, allowing us to determine the true function shift, $y_i^{FSL}$. We then train the model to regress this value, optimized with an $\mathcal{L}_1$ objective:

$$\min_\theta \; \mathbb{E}_{\mathcal{G} \sim \mathcal{D}}\left[\left|\psi(x_i) - y_i^{FSL}\right|\right], \tag{5}$$

where $x_i$ is the final embedding for node $i$ and $\psi(\cdot)$ is a 3-layer MLP regression head.

**Inference Stage.** At the inference stage, the true global functions are unknown. Since computing the global function of a node at any given level depends on the global functions of its predecessors from previous levels, we cannot predict them all at once. Therefore, we reconstruct the global functions iteratively, proceeding level by level through the circuit. For each node, we first compute its local function using the already-estimated global functions of its inputs. The final estimate for the node's global function is then obtained by adding the model's predicted function shift to this computed local function. This entire iterative process is detailed in Algorithm 1.

---

**Algorithm 1** Inference with Function Shift

**Input:** Circuit graph $\mathcal{G} = (V, E)$
1: $\mathbf{x} \leftarrow Hierarchical\_Transformer(\mathcal{G})$
2: $\hat{\mathbf{y}}^{FSL} \leftarrow \psi(\mathbf{x})$
3: $L \leftarrow \max_{v \in V} \text{level}(v)$
4: **for** $l = 1$ to $L$ **do**
5:     **for** $v \in \{u \in V : \text{level}(u) = l\}$ **do**
        $\hat{y}_v^{global} \leftarrow \hat{y}_v^{FSL} +$
6:         $\underbrace{\phi_v(\hat{y}_{u_1}^{global}, \ldots, \hat{y}_{u_{|\mathcal{N}(v)|}}^{global})}_{\text{Local Function}}$
7:     **end for**
8: **end for**
9: **Return** $\hat{\mathbf{y}}^{global}$ for all nodes

---

### 3.4 Contrastive Task

Contrastive learning is a standard self-supervised strategy for learning representations of circuit functionality (Wang et al., 2024; Fang et al., 2025b;c; Wu et al., 2025). The fundamental principle is

to pull embeddings of functionally equivalent circuits closer together in the embedding space while pushing apart those of functionally different circuits. This process trains the model to identify and encode the discriminative features that define a circuit's intrinsic properties, all without needing explicit labels. Following prior work, we form training instances for each circuit $\mathcal{G}$. A positive sample, $\mathcal{G}^+$, is created by applying a functionally equivalent transformation to $\mathcal{G}$. All other circuits within the same batch are treated as negative samples, $\mathcal{G}^-$. We then optimize the encoder using the InfoNCE loss (Oord et al., 2018):

$$\min_\theta \ \mathbb{E}_{\mathcal{G}\sim\mathcal{D}} \ \mathcal{L}_{\text{InfoNCE}}(\mathcal{G}, \mathcal{G}^+, \mathcal{G}^-). \tag{6}$$

The effect of this objective is to structure the embedding space such that functional equivalence corresponds to proximity, mapping similar circuits to nearby points while separating them from non-equivalent ones.

## 4 EXPERIMENT

### 4.1 IMPLEMENT DETAILS

**Dataset**   In this paper, we conduct experiments on three modalities: RTL, AIG and PM netlist. For RTL, we follow previous works (Fang et al., 2025b;c) and collect data from ITC (Corno et al., 2002) and OpenCores (Albrecht, 2005). For combinational AIGs, we use ForgeEDA (Shi et al., 2025a). For Sequential AIGs, we follow DeepSeq2 (Khan et al., 2025) and extract sub-circuits from ITC (Corno et al., 2002), OpenCores (Albrecht, 2005) and ISCAS'89 (Brglez et al., 1989). For PM netlist, we follow previous work (Shi et al., 2025b) and extract sub-circuits from ForgeEDA (Shi et al., 2025a) and DeepCircuitX (Li et al., 2025). More details are provided in Appendix C.

**Evaluation Metrics**   In this work, we evaluate our model on two types of tasks: contrastive and predictive. For the first type, the contrastive task, the goal is retrieval. Given a query circuit, the model must identify its functionally equivalent (positive) counterpart from a pool of $N$ candidate circuits. For our experiments, we set the pool size to $N = 256$ for RTL and $N = 1024$ for AIG and PM Netlists. We measure performance using the Recall@k (Rec@k) metric, reporting scores for $k \in \{1, 5, 10\}$. For the second type, the predictive tasks, we assess the model's ability to determine node-level properties. We follow previous works (Shi et al., 2023; Khan et al., 2025; Shi et al., 2025b) and perform logic-1 probability prediction, similarity prediction and transition probability prediction (See Appendix E). After encoding a circuit to produce node embeddings, these are used to predict a target value for each node. We evaluate the accuracy of these predictions using Mean Absolute Error (MAE) and the $R^2$ score.

Table 1: Comparison of contrastive task across various modalities(%).

| Model | RTL | | | AIG | | | Netlist | | |
|---|---|---|---|---|---|---|---|---|---|
| | Rec@1 | Rec@5 | Rec@10 | Rec@1 | Rec@5 | Rec@10 | Rec@1 | Rec@5 | Rec@10 |
| *Message Passing Neural Network* | | | | | | | | | |
| GCN | 82.90 | 87.43 | 91.57 | 83.01 | 93.28 | 96.05 | 58.03 | 77.65 | 85.24 |
| GraphSAGE | 86.46 | 92.87 | 95.43 | 88.55 | 96.41 | 98.38 | 86.12 | 95.82 | 98.13 |
| GAT | 84.98 | 89.22 | 94.65 | 85.68 | 94.32 | 97.60 | 65.21 | 83.72 | 89.97 |
| GIN | 86.23 | 91.34 | 96.95 | 85.98 | 93.40 | 96.52 | 75.85 | 91.82 | 95.90 |
| FGNN2 | - | - | - | 88.73 | 97.03 | 98.57 | - | - | - |
| DeepCell | - | - | - | - | - | - | 80.99 | 95.31 | 97.61 |
| *Graph Transformer* | | | | | | | | | |
| GraphGPS | 86.94 | 92.13 | 96.37 | OOM | OOM | OOM | 45.63 | 61.55 | 69.57 |
| SGFormer | 79.45 | 86.57 | 89.50 | 15.43 | 30.88 | 42.19 | 15.83 | 37.06 | 49.73 |
| DIFFormer | 88.28 | 92.97 | 96.88 | 37.03 | 68.87 | 80.66 | 25.23 | 45.52 | 55.67 |
| CircuitEncoder | 88.27 | 92.97 | 94.52 | - | - | - | - | - | - |
| **TRACE** | **94.45** | **98.74** | **99.89** | **92.68** | **98.65** | **99.51** | **90.81** | **98.48** | **99.44** |

## 4.2 Contrastive Tasks

**RTL** On the RTL modality, traditional message-passing models such as GCN (Kipf, 2016), GIN (Xu et al., 2018), GAT (Veličković et al., 2017), and GraphSAGE (Hamilton et al., 2017) achieve moderate performance, with Rec@1 ranging from 82.9% to 86.5%. Graph Transformer variants like GraphGPS (Rampášek et al., 2022), SGFormer (Wu et al., 2023c), DIFFormer (Wu et al., 2023b), and the RTL-specialized CircuitEncoder (Fang et al., 2025c) show competitive scores, with Rec@1 ranging from 79.45% to 88.28%. In contrast, TRACE achieves a substantial improvement, pushing Rec@1 to 94.45% and Rec@10 to 99.89%, outperforming the second-best method by 6.17% and 2.94% respectively.

**AIGs** For AIGs, the performance gap between baselines and TRACE becomes even more pronounced. While traditional message-passing networks perform decently (with Rec@1 from 83.01% to 88.55%) and AIG-specialized architectures like FGNN2 (Wang et al., 2024) reach 88.7%, Graph Transformer models generally struggle. For instance, performance of SGFormer and DIFFormer drop to as low as 15.4% and 37.03% Rec@1 respectively, highlighting the drawback of methods that destroy the explicit hierarchy in a computational graph. GraphGPS even suffers from an Out-of-Memory (OOM) error due to its dense attention mechanism. Our method, however, consistently outperforms all baselines, reaching 92.68% at Rec@1 and 99.51% at Rec@10.

**PM Netlists** The Netlist modality presents the most challenging benchmark, where message-passing models show varying performance. For instance, GraphSAGE and the netlist-specialized DeepCell (Shi et al., 2025b) achieve strong results (Rec@1 around 80–86%), while others like GCN and GAT show a significant performance drop. Graph Transformer models also suffer from severe performance degradation, with Rec@1 scores below 50% for GraphGPS, SGFormer, and DIFFormer. Remarkably, TRACE delivers consistent and superior performance, attaining 90.81% Rec@1 and 99.44% at Rec@10, outperforming both families of baselines by a large margin.

In summary, TRACE consistently outperforms both message-passing and Transformer-based baselines across all three modalities. Furthermore, its stable performance across these diverse graph types highlights its strong capacity for generalization.

Table 2: Comparison of predictive tasks on combinational and sequential AIGs.

| Model | Combinational AIG | | | | Sequential AIG | | | |
|---|---|---|---|---|---|---|---|---|
| | Logic-1 Probability | | Similarity Prediction | | Logic-1 Probability | | Transition Probability | |
| | $R^2$ | MAE | $R^2$ | MAE | $R^2$ | MAE | $R^2$ | MAE |
| *Message Passing Neural Network* | | | | | | | | |
| GCN | 0.644 | 0.152 | 0.271 | 0.090 | 0.868 | 0.064 | 0.744 | 0.024 |
| GAT | 0.618 | 0.157 | 0.029 | 0.090 | 0.877 | 0.053 | 0.831 | 0.016 |
| GIN | 0.669 | 0.144 | 0.445 | 0.076 | 0.962 | 0.035 | 0.790 | 0.023 |
| GraphSAGE | 0.675 | 0.143 | 0.438 | 0.078 | 0.927 | 0.048 | 0.867 | 0.017 |
| DeepGate2 | 0.983 | 0.028 | 0.502 | 0.069 | - | - | - | - |
| PolarGate | 0.493 | 0.192 | 0.021 | 0.113 | - | - | - | - |
| MGVGA | 0.666 | 0.145 | 0.418 | 0.077 | - | - | - | - |
| DeepSeq2 | - | - | - | - | 0.979 | 0.025 | 0.908 | 0.014 |
| *Graph Transformer* | | | | | | | | |
| GraphGPS | OOM | OOM | OOM | OOM | 0.971 | 0.026 | 0.901 | 0.012 |
| SGFormer | 0.516 | 0.175 | -0.072 | 0.117 | 0.878 | 0.056 | 0.596 | 0.026 |
| DIFFormer | OOM | OOM | OOM | OOM | 0.701 | 0.097 | 0.416 | 0.034 |
| DeepGate4 | 0.984 | 0.027 | 0.464 | 0.078 | - | - | - | - |
| **TRACE** | **0.989** | **0.015** | **0.633** | **0.055** | **0.997** | **0.009** | **0.976** | **0.005** |

## 4.3 Predictive Tasks

**Combinational AIGs** On combinational AIGs, message-passing baselines such as GCN, GIN, GAT, GraphSAGE, PolarGate (Liu et al., 2024) and MGVGA (Wu et al., 2025) yield poor performance, with $R^2$ values ranging from 0.493 to 0.675 for logic-1 probability prediction and from 0.021 to 0.445 for similarity prediction. Graph Transformer baselines either suffer from OOM or achieve limited performance. AIG-specialized architectures, like DeepGate2 (Shi et al., 2023),

DeepGate4 (Zheng et al., 2025) improves results with $R^2$ around 0.98 for logic-1 probability and around 0.50 for similarity. In contrast, TRACE delivers the best overall performance, achieving an $R^2$ of 0.989 with MAE 0.015 for logic-1 probability and significantly outperforming all baselines in similarity prediction with $R^2$ of 0.633 and MAE of 0.055.

**Sequential AIGs**  Sequential circuits pose additional challenges due to temporal dependencies. Our method again demonstrates substantial gains, reaching an $R^2$ of 0.997 with MAE 0.009 for logic-1 probability, and 0.976 with MAE 0.005 for transition probability, outperforming the second-best method, DeepSeq2 (Khan et al., 2025), by 0.018 in $R^2$ on logic-1 probability prediction and 0.068 in $R^2$ on transition probability prediction.

**PM Netlists**  The predictive task on PM netlists further validates the generalization ability of TRACE. Message-passing methods achieve moderate performance, with GraphSAGE and DeepCell reaching $R^2$ above 0.94. Graph Transformer models show mixed results, with SGFormer performing relatively well ($R^2 = 0.918$) but DIFFormer dropping to 0.696. TRACE clearly surpasses all baselines, achieving an $R^2$ of 0.994 and a minimal MAE of 0.013. This consistent superiority across modalities emphasizes the adaptability of our approach to different circuit representations and predictive objectives.

Overall, across all predictive tasks, TRACE not only surpasses both message-passing and Transformer-based models, but also approaches near-perfect accuracy, demonstrating its capacity to generalize across combinational, sequential, and physical design graph domains.

Table 3: Comparison of predictive task on PM netlists.

| Model | $R^2$ | MAE |
|---|---|---|
| *Message Passing Neural Network* | | |
| GCN | 0.718 | 0.112 |
| GraphSAGE | 0.946 | 0.048 |
| GAT | 0.902 | 0.059 |
| GIN | 0.734 | 0.102 |
| DeepCell | 0.942 | 0.053 |
| *Graph Transformer* | | |
| GraphGPS | 0.846 | 0.083 |
| SGFormer | 0.918 | 0.056 |
| DIFFormer | 0.696 | 0.141 |
| **TRACE** | **0.994** | **0.013** |

### 4.4 ABLATION STUDY

Table 4: Ablation study on function shift learning (FSL).

| Setting | PM Netlist | | AIG | | | |
|---|---|---|---|---|---|---|
| | Logic-1 Probability | | Logic-1 Probability | | Similarity Prediction | |
| | $R^2$ | MAE | $R^2$ | MAE | $R^2$ | MAE |
| TRACE w/o FSL | 0.985 | 0.036 | 0.980 | 0.024 | 0.500 | 0.066 |
| TRACE | **0.994** | **0.013** | **0.989** | **0.015** | **0.533** | **0.055** |

We conducted an ablation study to quantify the contribution of our proposed Function Shift Learning (FSL) component, which is designed to help our model adapt to different graph types and their unique functional distributions. The results, summarized in Table 4, demonstrate that FSL is a crucial element for achieving high performance. The model with FSL consistently outperforms its ablated counterpart (TRACE w/o FSL) across all tasks and datasets. For Logic-1 Probability prediction on PM Netlists, the addition of FSL significantly reduces the MAE from 0.036 to 0.013 and increases the $R^2$ score from 0.985 to 0.994, indicating more accurate predictions. As for AIGs, FSL improves the MAE by 0.009 on logic-1 probability prediction and 0.011 on similarity prediction. This confirms that the FSL component is essential for our model's ability to learn functional representations, leading to superior performance on diverse computational graphs.

## 5 CONCLUSION

In this work, we introduced TRACE, a new paradigm for learning on computational graphs that addresses the architectural limitations of conventional MPNNs and Transformers. By employing a novel Hierarchical Transformer and a function shift learning objective, TRACE directly models the position-aware, hierarchical nature of computation. Our extensive experiments on electronic circuits demonstrate that TRACE substantially outperforms all prior architectures, establishing a new state of the art. This work provides a proof of principle for a more architecturally sound approach to learning on computational graphs, offering a powerful framework with potential applications across various domains.

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

## A  THE USE OF LARGE LANGUAGE MODELS

In the preparation of this paper, we utilized a large language model (LLM) as an assistive tool to enhance the quality of our writing and presentation. The LLM's role was strictly confined to refining the manuscript's writing and formatting, without generating any core scientific content or data.

## B  COMPUTATION GRAPH

**Register Transfer Level(RTL)**   RTL design is a hardware abstraction used in the early stages of digital chip design, serving as the bridge between high-level behavioral descriptions and gate-level implementations. To be specific, RTL code captures how data moves between registers (i.e., sequential registers) and how logic gates operate on that data within each clock cycle (i.e., combinational logic). Essentially, an RTL design can be viewed as a directed graph. We first convert the RTL code in Hardware Description Language (HDL) format into an abstract syntax tree (AST) and then extract the graph structure based on this tree. In this graph, nodes represent word-level register signals and various operators (e.g., And, Add, Equal, Mux), while the wires in the HDL code form the edges that denote the paths of data flow.

**And-Inverter Graph(AIG)**   In our work, we use combinational AIG and sequential AIG, which are widely used for circuit analysis, optimization, and formal verification because of their compact and canonical representation of Boolean functions.
Combinational AIG is a directed acyclic graph (DAG) composed of three basic elements: Primary Input(PI), AND gate and NOT gate. Since any Boolean logic expression can be constructed using only AND and NOT operations, AIG provides a universal and efficient representation. For example, a simple logic expression $\neg A \wedge B$ can be built as a DAG with 2 PIs(A and B), one NOT gate and one AND gate. The edges are $[(A, NOT), (NOT, AND), (B, AND)]$. The AND gate with no outgoing edges represents the circuit's final output in this DAG.
Sequential AIG extends this by introducing registers as an additional node type. These registers can capture the circuit's state at each clock cycle, enabling sequential AIG to represent more complex circuits with memory functionality, such as finite state machines.

**Post-Mapping Netlist**   A Post-Mapping Netlist is a gate-level representation obtained after logic synthesis and technology mapping, where the circuit is expressed using standard cells from a target technology library and optimized for timing, area, and power. Unlike AIGs, which represent circuits as abstract DAGs of AND and NOT gates (and optionally registers for sequential circuits) focusing on low-level function, post-mapping netlists capture high-level implementation details, including specific gate types and connectivity imposed by the target library. Consequently, the node types and structures can differ significantly from those in AIGs, which is why we treat AIGs and Post-Mapping (PM) netlists as distinct modalities in this paper.

## C  DATASET DETAILS

We summarize the statistics of the datasets used in both contrastive tasks (Table 5) and predictive tasks (Table 6). For each dataset, we report the number of graphs (#Graphs), the number of nodes (#Nodes), the number of edges (#Edges), and the maximum depth of a graph (Depth). For #Nodes, #Edges, and Depth, we provide the minimum, average, and maximum values, denoted as $\{min, avg, max\}$.

Table 5: Dataset statistics for contrastive tasks with $\{min., avg., max.\}$.

|  | RTL | AIG | PM Netlist |
|---|---|---|---|
| #Graphs | 1138 | 67706 | 67728 |
| #Nodes | { 9.0, 102.6, 1987.0} | {22.0, 162.2, 2127.0} | {16.0, 88.6, 1192.0} |
| #Edges | {10.0, 278.3, 2645.0 } | {22.0, 176.0, 2275.0} | {20.0, 110.5, 1361.0} |
| Depth | {2.0, 8.9, 27.0} | {6.0, 16.7, 29.0} | {2.0, 5.9, 12.0} |

Table 6: Dataset statistics for predictive tasks with $\{min., avg., max.\}$.

| | Com. AIG | Seq. AIG | PM Netlist |
|---|---|---|---|
| #Graphs | 9800 | 10007 | 83042 |
| #Nodes | $\{10.0, 2324.9, 45409.0\}$ | $\{23.0, 236.5, 1281.0\}$ | $\{12.0, 668.1, 4783.0\}$ |
| #Edges | $\{12.0, 3268.57, 68676.0\}$ | $\{21.0, 260.5, 1915.0\}$ | $\{11.0, 1113.9, 8914.0\}$ |
| Depth | $\{4.0, 49.1, 2657.0\}$ | $\{4.0, 21.0, 102.0\}$ | $\{1.0, 16.0, 260.0\}$ |

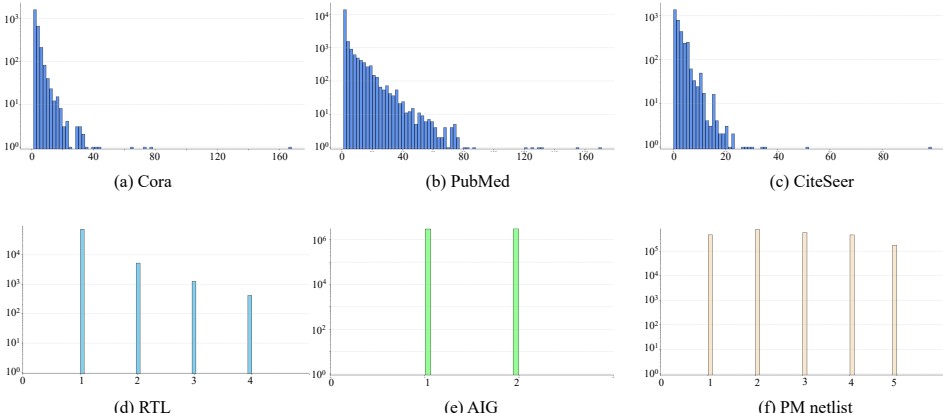

Figure 4: In-degree distribution. The x-axis represents the node in-degree and the y-axis represents the frequency (number of nodes).

## D  DEGREE DISTRIBUTION

In this section, we compare the in-degree properties of computational graphs and general graphs, and analyze the corresponding padding overhead induced by our method. Specifically, we study three types of computational graphs (RTL, AIG, and PM netlist) and compare them with citation graphs, a representative class of DAGs. For the latter, we use the Cora, CiteSeer, and PubMed datasets from Yang et al. (2016).

As shown in Figure 4, the in-degree of a node in a computational graph is largely determined by its operator type, leading to a stable and bounded distribution across RTL, AIG, and PM netlist. In contrast, general DAGs such as citation networks exhibit long-tailed in-degree distributions. This discrepancy is critical for our method (Section 3.2), as sequence length is padded according to node in-degree. Consequently, long-tailed distributions introduce significant redundancy for general DAGs, while computational graphs remain more compact.

We quantify the redundancy using the following metric:

$$\text{padding\_overhead} = \frac{n \times \max_i(d_i) - \sum_i d_i}{n \times \max_i(d_i)}, \tag{7}$$

where $n$ denotes the number of nodes in the graph and $d_i$ is the in-degree of node $i$.

Table 7: Padding Overhead Across Different Graphs

| Metric | Computational Graph | | | General Graph | | |
|---|---|---|---|---|---|---|
| | RTL | AIG | PM Netlist | Cora | CiteSeer | PubMed |
| Padding Overhead | 42.35% | 16.29% | 38.54% | 97.10% | 96.22% | 96.80% |

As summarized in Table 7, citation graphs suffer from severe padding overhead, ranging from 96.22% to 97.10%, which corresponds to nearly $20\times$ additional computation cost. In contrast, computational graphs exhibit much lower overhead, between 16.29% and 42.35%. These results

underscore the structural advantage of computational graphs: their bounded in-degree leads to substantially reduced padding, thereby improving the efficiency of our proposed method.

# E  PREDICTIVE TASKS

## E.1  LOGIC-1 PROBABILITY PREDICTION

Logic-1 probability prediction is a node-level regression task. In a digital circuit, the logic value of any node $i$ can be modeled as a binary random variable $x_i \sim \mathcal{B}(p_i)$, where $p_i$ is the probability of the node being in the logic "1" state. This value, often referred to as the signal probability, is a crucial indicator of circuit function. The objective of this task is to predict the parameter $p_i$ for each node in the circuit, providing insight into its functional behavior.

## E.2  FUNCTIONAL SIMILARITY PREDICTION

The core objective of similarity prediction is to predict the functional similarity between a given pair of nodes. To establish the ground truth for this task, we first sample a fixed set of input patterns from the complete input space (exhaustive simulation would require testing all $2^n$ possible input combinations for $n$ inputs). We sample a set of node pairs $\mathcal{N}_{\text{pairs}}$. For each node $i$ in a selected pair, we generate a partial truth table $T_i$ by recording its state under this shared set of input patterns. The functional similarity $S_{(i,j)}$ for a pair of nodes $(i, j) \in \mathcal{N}_{\text{pairs}}$ is then calculated based on the normalized Hamming distance between their respective partial truth tables, $T_i$ and $T_j$:

$$S_{(i,j)} = 1 - \frac{\text{HammingDistance}(T_i, T_j)}{\text{length}(T_i)} \tag{8}$$

This similarity score $S_{(i,j)}$ ranges from 0 to 1, where 1 indicates that the two nodes have identical outputs for all simulated input vectors, and 0 indicates they are completely dissimilar.

## E.3  TRANSITION PROBABILITY PREDICTION

To analyze the dynamic behavior of sequential circuits, random binary input sequences are applied to each primary input, and the circuit is simulated. For each input sequence, the output states $s_i(t) \in \{0, 1\}$ of standard cells and the sequential outputs of registers are recorded. The number of transitions from 0 to 1 and from 1 to 0 for each cell or register $i$ is counted as $N_{0 \to 1}^{(i)}$ and $N_{1 \to 0}^{(i)}$, respectively. The transition probabilities are then defined as

$$P_{0 \to 1}^{(i)} = \frac{N_{0 \to 1}^{(i)}}{N_{\text{total}}}, \quad P_{1 \to 0}^{(i)} = \frac{N_{1 \to 0}^{(i)}}{N_{\text{total}}}, \tag{9}$$

where $N_{\text{total}}$ is the total number of input sequences.

Notably, the output of each register changes according to the input sequence and clock cycles, and the initial value of each register in a simulation step is taken from the output of the previous simulation. These transitions reflect the activity of circuit nodes, which is the primary source of dynamic power consumption. Therefore, transition probabilities provide a quantitative measure of the dynamic switching characteristics of the circuit, and can be used as an indicator for dynamic power analysis.

