# OpenReview forum: "TRACE: Learning to Compute on Graphs"
_ICLR.cc/2026/Conference — ICLR 2026 Conference Withdrawn Submission_

### Official Review · Reviewer_QkfA · 2025-10-26

**Soundness:** 3
**Presentation:** 3
**Contribution:** 3
**Rating:** 4
**Confidence:** 4

**Summary:**

The authors argue that dominant models, particularly message passing neural networks (MPNNs), are architecturally mismatched for this task. This is because their core "permutation-invariant aggregation" mechanism makes it impossible to model the position-aware and hierarchical nature of computation (e.g., distinguishing between MUX(S, A, B) and MUX(A, S, B)). TRACE addresses this fundamental flaw with a two-fold approach:

Hierarchical Transformer and Function Shift Learning (FSL).

The method is validated on a comprehensive suite of electronic circuit benchmarks (RTL, AIG, and PM netlists). The results show that TRACE substantially outperforms all prior MPNN and Transformer-based architectures on both contrastive (retrieval) and predictive (node-level regression) tasks.

**Strengths:**

Clear Problem Definition: The paper's primary strength is its clear and compelling articulation of the fundamental architectural mismatch between MPNNs and computational graphs. The use of the MUX operator as a motivating example (Figure 1) is extremely effective and intuitive, immediately establishing why permutation-invariance is a critical failure point for this domain.

Architecturally Sound Solution: The proposed Hierarchical Transformer is an elegant and architecturally sound solution that directly addresses the identified problem. By processing nodes topologically (level by level) and using a Transformer to model each computation as an ordered sequence, the model's structure faithfully mirrors the actual data flow and dependencies of computation.

**Weaknesses:**

Limited Generality of Experiments: The paper is titled "Learning to Compute on Graphs", but the experiments are focused exclusively on electronic circuits. While circuits are an excellent and challenging example, it is unclear how well this paradigm generalizes to other types of computational graphs (e.g., software control data flow graphs, or the computation graphs of neural networks themselves). These graphs may have different properties, such as much larger and more varied in-degrees, which are not explored. If focusing on circuits, it's not proper with the current title.

Scalability Concerns (Graph Depth): The Hierarchical Transformer processes the graph in topological order, level by level, which is an inherently sequential process. The paper does not provide a detailed analysis of the method's scalability, particularly with respect to graph depth. A computational graph with a very long critical path (many sequential logic levels) might incur significant inference latency compared to more parallelizable MPNN architectures.

**Questions:**

On Generality: Following on the weaknesses, could the authors comment on the applicability of TRACE to non-circuit computational graphs? Specifically, how would the method handle graphs with the long-tailed in-degree distributions common in general graphs (as shown in Appendix D, Fig. 4)? This would lead to highly variable sequence lengths and significant padding overhead. Or one may change the title that focus on circuits.

On Scalability: What is the computational complexity of TRACE with respect to the number of nodes ($N$), edges ($E$), and the graph's logical depth ($L$)? How does its training and inference time scale on very deep graphs compared to asynchronous MPNNs, which also follow a topological order?

On Function Shift Learning (FSL): The FSL objective relies on pre-computing the ground-truth "global function" for supervision during training. How would this objective be applied in domains where this ground truth is intractable or impossible to compute or sample (e.g., for a complex software program)?

---

### Official Review · Reviewer_Ua4B · 2025-10-29

**Soundness:** 3
**Presentation:** 2
**Contribution:** 2
**Rating:** 4
**Confidence:** 3

**Summary:**

This paper introduces TRACE, a Hierarchical Transformer architecture for learning functional representations of computational graphs, with a focus on electronic circuits. The authors identify fundamental limitations of existing approaches: MPNNs use permutation-invariant aggregation that cannot model position-aware operators, while standard Graph Transformers either destroy hierarchical structure or reduce to MPNN-like mechanisms. TRACE addresses these issues through two innovations: (1) a Hierarchical Transformer that processes nodes level-by-level as ordered prefix notation sequences with positional encodings, enabling position-aware and operator-specific interactions among inputs, and (2) function shift learning (FSL), which trains models to predict the discrepancy between local (independence-assuming) and global (reconvergence-aware) functions rather than directly predicting complex global functions. Experiments across three circuit modalities (RTL, AIG, PM netlists) on both contrastive retrieval and predictive tasks demonstrate substantial improvements over MPNNs and Graph Transformer baselines.

**Strengths:**

1. **Clear problem formulation**. The paper provides compelling motivation for the Hierarchical Transformer design through concrete examples. Figure 1 effectively illustrates how MPNNs fundamentally fail on position-aware operators like MUX(S,A,B), where permutation-invariant aggregation cannot distinguish input orderings. The prefix notation representation with positional encodings elegantly addresses this limitation while maintaining computational efficiency through the bounded in-degree property of circuits (Appendix D shows only 16-42% padding overhead vs 96%+ for general graphs).

2. **Comprehensive experimental validation across diverse settings**. The evaluation is thorough and convincing, spanning multiple circuit modalities (RTL, AIG, PM netlists), both combinational and sequential circuits, and diverse tasks including contrastive retrieval and three predictive tasks (logic-1 probability, similarity, transition probability).

**Weaknesses:**

1. **Limited empirical analysis of function shift learning**. While FSL is presented as a key contribution for capturing reconvergent dependencies, the ablation study in Table 4 shows relatively modest improvements. On PM Netlist logic-1 probability prediction, R^2 improves from 0.985 to 0.994 (MAE from 0.036 to 0.013), and on AIG similarity prediction, R^2 increases from 0.500 to 0.533. The paper claims in lines 292-298 that FSL "decouples the global function" and "isolates the complex contextual effects caused by reconvergence," but provides limited empirical evidence demonstrating this decoupling actually occurs or that the model learns meaningful representations of function shifts.

2. **Narrow experimental scope limits claims about general computational graphs**. The paper frames the contribution broadly as addressing "the fundamental challenge of learning to compute on graphs" (lines 27, 108-109) and "learning on computational graphs" (abstract, title, conclusion), but evaluates exclusively on electronic circuits. The approach makes strong domain-specific assumptions: bounded in-degree distributions (Appendix D), existence of clear topological ordering for level-based processing (Equation 2), and specific operator types.

3. **Missing computational efficiency and scalability analysis**. The paper provides no concrete wall-clock time comparisons, memory consumption measurements, or empirical scalability analysis despite claiming efficiency advantages. While Appendix D analyzes padding overhead theoretically (16.29-42.35% for computational graphs), this doesn't directly translate to practical speedups because the hierarchical processing in Algorithm 1 requires sequential level-by-level passes rather than the parallel processing possible with synchronous MPNNs.

**Questions:**

1.  Could you provide more detailed empirical evidence that FSL actually learns to model reconvergent dependencies as claimed?

2.  Can TRACE generalize to other computational graphs mentioned in your introduction, such as control data flow graphs (CDFGs) in software engineering or dataflow graphs in machine learning frameworks? Have you conducted any experiments beyond circuits?

3. What are the actual wall-clock training and inference times compared to synchronous MPNNs and Graph Transformers on identical hardware?

---

### Official Review · Reviewer_ghHe · 2025-10-31

**Soundness:** 2
**Presentation:** 3
**Contribution:** 2
**Rating:** 2
**Confidence:** 4

**Summary:**

Most existing circuit models rely on asynchronous message passing that follows circuit topological order, which effectively simulates logic propagation but fail to capture the asymmetric role of nodes in the computational graphs (i.e., some operators are not permutation invariant).
This paper introduces TRACE that replaces the local message aggregation with attention-based aggregation along with positional encodings, thereby enabling to model operators with asymmetric interactions. It also proposes a better supervised learning loss called Function Shift Learning: to predict function $E_{x_1,...,x_n}\phi(x_1,...,x_n)$, we supervise the model to predict the difference $E_{x_1,...,x_n}\phi(x_1,...,x_n)-\phi(E_{x_1}x_1, ...,E_{x_n}x_n)$. Experiments across RTL, AIG, and layout-level benchmarks show that TRACE achieves consistent improvements over both standard GNNs and circuit-specific baselines.

**Strengths:**

- The limitations of standard asynchronous message passing are clearly explained and well motivated.
- The experimental results show consistent improvements across multiple circuit tasks compared to baselines, with ablation study showing the effectiveness of the proposed Function Shift Learning objective.

**Weaknesses:**

- The technical novelty is limited: TRACE mainly replaces local message aggregation with attention within an existing asynchronous framework. The so-called hierarchical aspect mainly reflects the existing asynchronous aggregation structure rather than a fundamentally new hierarchical mechanism.
- There is not ablation study on the role of positional encodings in attention. It is unclear how much performance the positional encodings in fact bring.
- Function Shift Learning (FSL) appears to be an effective training and inference pipeline for capturing global circuit behaviors. However, it is unclear whether FSL is exclusive for hierarchical transformers or can also benefit other circuit-learning models. Evaluating existing baselines under the same FSL framework would help clarify how much of the performance gain originates from hierarchical transformers itself versus the FSL pipeline.

**Questions:**

- Can Function Shift Learning (FSL) be applied to other circuit neural networks (e.g., DeepGate or FGNN2)? If so, would these models also benefit from the same FSL pipeline? If not, the authors are encouraged to clarify why FSL is particularly suited to TRACE.

---

### Official Review · Reviewer_xU9p · 2025-10-31

**Soundness:** 3
**Presentation:** 3
**Contribution:** 3
**Rating:** 6
**Confidence:** 3

**Summary:**

The paper argues computation on graphs demands position-aware, hierarchical processing and proposes TRACE, a level-wise hierarchical Transformer that encodes each operator as an ordered sequence of tokens to model multi-input interactions faithfully. A “function shift learning” objective supervises the discrepancy between a local independence-based approximation and the true global function, disentangling intrinsic operator behavior from reconvergent topology effects. Across RTL, AIG, and post-mapping netlists, TRACE achieves strong retrieval (e.g., high Rec@1) and accurate node-level predictions, outperforming MPNNs and graph transformers that rely on permutation-invariant aggregation or masked attention

**Strengths:**

1. Persuasive critique of MPNN permutation-invariance and hierarchy-agnostic transformers for computation, with operator-specific examples like MUX.
  2. Architecturally faithful backbone and a principled objective disentangling local vs.\ global functions.
3.  Broad, consistent empirical gains across multiple standard circuit suites.

**Weaknesses:**

1. Some claims about fundamental mismatch feel broad; consider scoping against newer graph-transformer variants.
2.  More ablations are needed to isolate contributions of the backbone vs.\ function shift learning.
3. Limited evidence of generality beyond circuits and specialized computational DAGs.

**Questions:**

1.  Sensitivity to the local approximation underlying function shift learning and potential failure modes.
2. Handling of sequential feedback and resource implications beyond pseudo-PI treatment.
3. Report compute trade-offs vs.\ operator-aware graph transformers with strong positional encodings.

---

### Note · Authors · 2025-11-16

I have read and agree with the venue's withdrawal policy on behalf of myself and my co-authors.